# Multisensory stimuli enhance the effectiveness of equivalence learning in healthy children and adolescents

**Gabriella Eördegh**[1], **Kálmán Tót**[2], **Ádám Kiss**[2], **Szabolcs Kéri**[2], **Gábor Braunitzer**[3], **Attila Nagy**[2]*

**1** Faculty of Health Sciences and Social Studies, University of Szeged, Szeged, Hungary, **2** Department of Physiology, Faculty of Medicine, University of Szeged, Szeged, Hungary, **3** Nyírő Gyula Hospital, Laboratory for Perception & Cognition and Clinical Neuroscience, Budapest, Hungary

* nagy.attila.1@med.u-szeged.hu

**Data Availability Statement:** All relevant data will be within the manuscript and its Supporting Information files.

## Abstract

It has been demonstrated earlier in healthy adult volunteers that visually and multisensory (audiovisual) guided equivalence learning are similarly effective. Thus, these processes seem to be independent of stimulus modality. The question arises as to whether this phenomenon can be observed also healthy children and adolescents. To assess this, visual and audiovisual equivalence learning was tested in 157 healthy participants younger than 18 years of age, in both a visual and an audiovisual paradigm consisting of acquisition, retrieval and generalization phases. Performance during the acquisition phase (building of associations), was significantly better in the multisensory paradigm, but there was no difference between the reaction times (RTs). Performance during the retrieval phase (where the previously learned associations are tested) was also significantly better in the multisensory paradigm, and RTs were significantly shorter. On the other hand, transfer (generalization) performance (where hitherto not learned but predictable associations are tested) was not significantly enhanced in the multisensory paradigm, while RTs were somewhat shorter. Linear regression analysis revealed that all the studied psychophysical parameters in both paradigms showed significant correlation with the age of the participants. Audiovisual stimulation enhanced acquisition and retrieval as compared to visual stimulation only, regardless of whether the subjects were above or below 12 years of age. Our results demonstrate that multisensory stimuli significantly enhance association learning and retrieval in the context of sensory guided equivalence learning in healthy children and adolescents. However, the audiovisual gain was significantly higher in the cohort below 12 years of age, which suggests that audiovisually guided equivalence learning is still in development in childhood.

## Introduction

Equivalence learning is a specific kind of associative learning in which two discrete and often different percepts are linked together. Catherine E. Myers and coworkers developed a learning

**Funding:** AN: SZTE ÁOK-KKA Grant No.:2019/270-62-2 Funder: University of Szeged, Faculty of Medicine, Szeged, Hungary http://www.med.u-szeged.hu/karunkrol/kari-palyazatok/aok-kari-kutatasi-alap-181005 The funders had no role in study design, data collection and analysis, decision to publish, or preparation of the manuscript. KT: EFOP 3.6.3-VEKOP-16-2017-00009 Funder: Goverment of Hungary https://www.palyazat.gov.hu/efop-363-vekop-16-felsoktatsi-hallgatk-tudomnyos-mhelyeinek-s-programjainak-tmogatsa# The funders had no role in study design, data collection and analysis, decision to publish, or preparation of the manuscript.

**Competing interests:** The authors have declared that no competing interests exist.

paradigm (the Rutgers Acquired Equivalence Test, also known as the fish-face paradigm) that can be applied to investigate visually guided equivalence learning [1]. A significant advantage of this test is that the brain regions associated with successful performance in each phase of the test are well established [1, 2]. The test can be divided into two main phases. The first one is the acquisition phase, which depends on the fronto-striatal [3] (cortex-basal ganglia) loops. Here the participants' task is to associate two different visual stimuli based on feedback information about the correctness of the choices. After the acquisition phase, once the participants have learned the associations, the test phase ensues. The test phase assesses memory retrieval regarding the learned associations (retrieval) and also tests if the subject is able to generalize from the known associations- that is, to recognize hitherto not seen but predictable stimulus pairs (generalization or transfer). During the test phase, which primarily depends on the hippocampi and the mediotemporal lobes [3], no feedback is given about the correctness of the choices.

Earlier studies have pointed out that both the basal ganglia and the hippocampi are fundamentally involved in visual associative learning [1–3], and they receive not only visual but also multisensory information [4–7]. Multisensory integration can be observed from the cellular to the behavioral level [5, 8–11]. To explore whether multisensory (audiovisual) information could facilitate the effectiveness of sensory guided equivalence learning, we developed and validated a new multisensory (audiovisual) equivalence learning test with the same structure as the original (visual) Rutgers Acquired Equivalence test [12, 13]. In a previous study involving 151 healthy adult volunteers, we demonstrated that visual and multisensory guided associative learning are similarly effective. Thus, these processes are independent of stimulus modality in healthy adults, but it is not known if the same applies to children and adolescents.

Concerning the development of multisensory integration in childhood, the available data are controversial and they strongly depend on stimulus modality and the studies cognitive function. The literature distinguishes between two main types of multisensory integration: the integration of different modalities and the integration of redundant stimulus features (e.g., spatial or temporal integration). The integration of different modalities is not detectable until 8 to 10 years of age in the auditory and tactile modalities [14, 15], and audiovisual integration is suboptimal (but detectable) until 11 to 12 years of age [16–21]. Therefore, in this study we also sought to investigate if there was a difference in participants' performance depending on whether they were above or below 12 years of age.

## Materials and methods

### Subjects

Altogether 167 healthy children and adolescents were involved in the study. The participants were recruited on a voluntary basis, received no compensation for their participation, and they were free to quit at any time without any consequence. The volunteers and their parents were informed about the aims and procedures of the study, and their medical history was taken with emphasis on neurological, ontological, psychiatric or chronic somatic disorders. Volunteers with such disorders in their history were not eligible for the study. Any regularly taken medication was recorded. The volunteers were also tested with the Ishihara plates to exclude color blindness. As all volunteers were under 18 years of age, the informed consent form was signed by their parents for them as required by the law. All volunteers were White and they were all native speakers of the Hungarian language. The study protocol followed the tenets of the Declaration of Helsinki in all respects, and it was approved by the Ministry of Human Resources (11818-6/2017/EÜIG).

## Visual and multisensory associative learning paradigms

The tests were administered on laptops (Lenovo T430, Lenovo Yoga Y500, Samsung Electronics 300e4z/300e5z/300e7z, Fujitsu Siemens Amilo Pro V3505). The subjects were tested in a quiet room, sitting at a standard distance of 57cm from the laptop screen (the stimuli were equal in size, with a maximum diameter of 5 cm, which corresponds to a 5° angle of view). For the audiovisual test, Sennheiser HD439 over-ear headphones were used to generate the auditory stimuli (SPL = 60 dB). The keys X and M were labeled as "left" and "right" on the laptop's keyboard. The subjects used these keys to indicate their choices in both test paradigms. The participants used both hands for the responses. The subjects were tested separately, one subject at a time. No time limit was set, and no forced quick responses were expected.

Both paradigms consisted of two phases: the acquisition phase and the test phase. The test phase could be further divided into two parts: a retrieval part and a generalization (or transfer) part. During the acquisition phase, the subjects had to learn associations between antecedent and consequent stimuli. This happened through trial-and-error learning. In each trial, one of two consequent stimuli had to be chosen in response to an antecedent stimulus. The subjects indicated their choice by pressing either the "left" or the "right" key on the keyboard, corresponding to the side of the consequent stimulus. The computer provided feedback about the correctness of the response–a green checkmark if the response was correct or a red X if it was incorrect, along with the Hungarian words "helyes" (correct) and "helytelen" (incorrect) (Fig 1).

New associations were presented one by one, and the participants had to provide a certain number of correct responses (4,6,8,10,12) after each new association before being allowed to proceed to the test phase. Thus, the number of trials was not constant in the acquisition phase; it depended on the subjects' individual performance.

In the test phase, the subjects first had to retrieve the already learned associations (the retrieval part of the test phase) then recognize new, hitherto not learned but predictable associations (generalization or transfer part of the test phase). These new associations were generated according to the previously formed associations that had been applied in the acquisition phase. In the test phase, no feedback was provided about the correctness of the answers. The number of trials was constant in the test phase. A total of 48 trials were presented, of which 36 were already learned (retrieval), and 12 were new associations (generalization or transfer).

The basis of the applied visual associative test was the Rutgers Acquired Equivalence Test [1]. It was rewritten in Assembly for Windows, translated to Hungarian, and slightly modified (more trials in test phases to get more accurate information about the hippocampal functions) [22] with the written permission of professor Catherine E. Myers (Rutgers University), head of the research group where the test paradigm was originally developed. The antecedent visual stimuli were four cartoon faces (an adult man, an adult woman, a boy, and a girl; A1, A2, B1, B2), and the consequents were four cartoon sematic fish of different colors but of the same shape (X1, X2, Y1, Y2). It was possible to form altogether eight pairs from the antecedent and consequent stimuli. In each trial (See Fig 1), the subjects saw a face in the middle of the screen and two fish below it, one on the left and one on the right side. During the acquisition phase, the subjects learned a series of antecedent-consequent pairs in a trial-and-error manner. When face A1 or face A2 were shown, the correct choice was fish X1 over fish Y1; however, when face B1 or face B2 appeared on the screen, the correct answer was fish Y1, instead of fish X1. This way, beside the face-fish associations, the participants also learned that the face A1 was equivalent to face A2 in terms of their relation to the consequents (fish). New associations were introduced gradually, and they were presented mixed with trials of previously learned associations until six of the possible eight antecedent–consequent pairs were encountered by

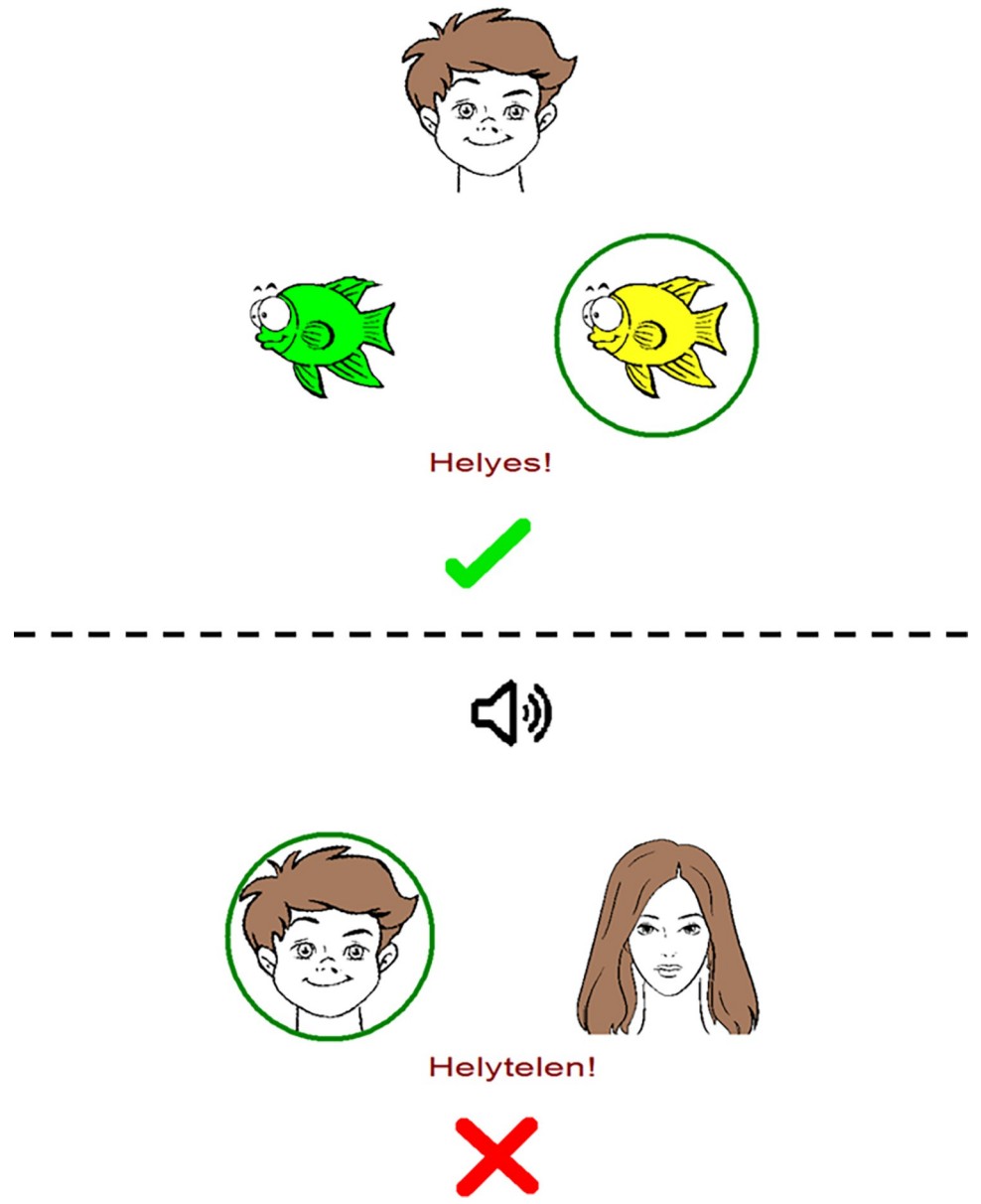

**Fig 1.** One trial from the visual (top) and the audiovisual (bottom) paradigms. Computer feedback to correct and incorrect responses (top and bottom, respectively) is also illustrated.

the participants. In the test phase, the participants had to recall these six pairs (retrieval), and the remaining two hitherto not presented combinations would be shown as well (generalization or transfer). If the participants successfully learned that A1 and A2 (or B1 and B2) were equivalent regarding their consequents, they could derive the rule and generalize it to make previously not learned associations. That is, by generalization, they inferred that consequent X2 (previously associated with antecedent A1) was also associated with antecedent A2 and consequent Y2 (previously associated with antecedent B1) was also associated with antecedent B2. These new associations were mixed with the old ones and the subjects were not informed about them.

The structure of the audiovisual paradigm was the same as that of the visual paradigm; the only difference was that the subjects had to make associations between auditory (antecedent stimuli, A1, A2, B1, B2) and visual stimuli (consequents, X1, X2, Y1, Y2) [12]. The antecedent stimuli were clearly distinguishable sounds (cat's meow, starting motor, guitar note, and woman saying a Hungarian word), and the consequents were the same four drawn faces as in the visual paradigm (adult man, adult woman, boy, and girl; A1, A2, B1, B2). In each trial, the subjects simultaneously heard a sound (SPL = 60 dB) through a loudspeaker and saw two faces on the right and left sides of the screen.

The participants had to learn which face was associated with which sound. Table 1 summarizes the basic structure of the learning tests.

The subjects completed both equivalence learning tests one after another. To avoid the carry-over effect, the tests were administered in a random order across the subjects.

## Data analysis

The performance of the participants was characterized with four main parameters: the number of trials necessary for the completion of the acquisition phase (NAT), association learning error ratio (the ratio of incorrect choices during the acquisition trials, ALER), retrieval error ratio (RER), and generalization error ratio (GER). Error ratios were calculated by dividing the number of incorrect responses by the total number of guesses. Reaction times were recorded for ALER, RER and GER. Reaction times (RTs) defined as the time elapsed between the appearance of the stimuli and the subject's response were also recorded for each trial. RT values over 3 SD of each participant's individual average RT were excluded from further analysis.

Statistical analysis was performed in Statistica 13.4.0.14 (TIBCO Software Inc., USA). NAT, ALER, RER and GER were compared between the visual and the audiovisual paradigms. As the data were non-normally distributed (Shapiro-Wilk $p < 0.05$), the Wilcoxon matched-pairs test was used for the hypothesis tests. We also analyzed multisensory gain and its correlation with the subjects' age. Gain was defined as the difference in the performance values between the visual (V) and multisensory (M) paradigms. For example: GAIN NAT = MNAT—VNAT. For the correlation analysis, Spearman's $\rho$ was calculated. Multisensory gain was also compared between the cohorts. For this, the Mann-Whitney U test was used.

## Results

Altogether167 healthy children and adolescents participated in the study. In three cases, due to technical reasons, the procedure was stopped. Four participants did not complete any of the two

**Table 1. Summary of the visual and audiovisual associative learning paradigms.**

| ACQUISITION | | | TEST | |
|---|---|---|---|---|
| **Shaping** | **Equivalence training** | **New consequents** | **Retrieval** | **Generalization** |
| A1 -> X1 | A1 -> X1 | A1 -> X1 | A1 -> X1 | |
| | A2 -> X1 | A2 -> X1 | A2 -> X1 | |
| | | A1 -> X2 | A1 -> X2 | |
| | | | | A2 -> X2 |
| B1 -> Y1 | B1 -> Y1 | B1 -> Y1 | B1 -> Y1 | |
| | B2 -> Y1 | B2 -> Y1 | B2 -> Y1 | |
| | | B1 -> Y2 | B1 -> Y2 | |
| | | | | B2 -> Y2 |

A, B: antecedents (faces in the visual and sounds in the audiovisual paradigm); X, Y: consequents (fish in the visual and faces in the audiovisual paradigm). For a detailed description, see text.

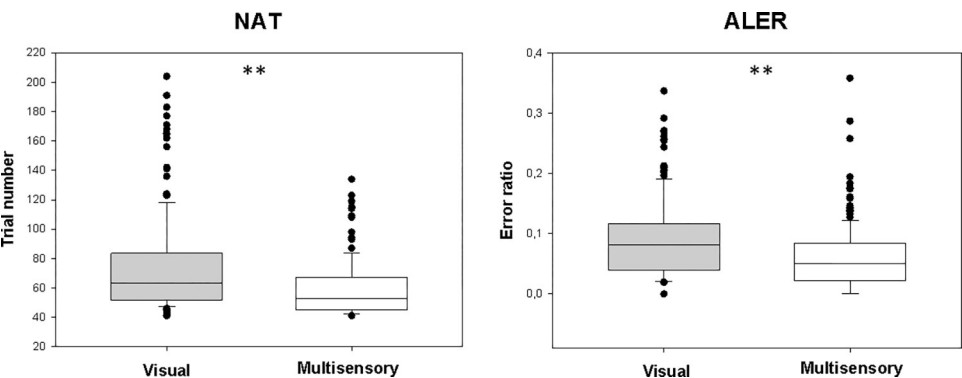

**Fig 2. Performance in the acquisition phase in the visual and audiovisual paradigms.** NAT: the number of trials needed to complete the acquisition phase; ALER: error ratio in the acquisition phase. Gray: visual; white: audiovisual. The lower margin marks the first quartile and the upper margin the third quartile. The line in the box marks the median. The whiskers below the boxes indicate the 10th percentile and the whiskers above, the 90th percentile. The black dots represent the outliers. **: p<0.01.

paradigms, and three could complete only the visual paradigm. Six percent (10/167) of the participants did not complete the procedure. Their data were not used in the analyses. This way, the data of 157 volunteers were analyzed ($n_{male}$ = 65, age: 11.6±3.6 years, range: 5–17.5 years).

## Comparison between the performances in the visual and multisensory learning paradigms

The median NAT in the visual paradigm was 63 (range: 41–204, n = 157), while in the audiovisual paradigm it was 53.0 (range: 41–134, n = 157). The median NAT in the audiovisual paradigm was significantly lower (Z = 5.098, p < 0.001; Fig 2).

The median ALER in the visual paradigm was 0.082 (range: 0–0.34, n = 157), and it was 0.051 in the multisensory paradigm (range: 0–0.36, n = 157). Similarly, to the NATs, the ALERs differed significantly between the two paradigms (Z = 4.652, p < 0.001; Fig 2).

In contrast to the psychophysical parameters, the RTs did not differ significantly between the two paradigms (Z = 0.050, p = 0.960) in the acquisition phase (AcqRTs). The median RT in the visual paradigm was 1655.811 ms (range: 885.508–4782.44ms, n = 157), and it was 1695.7 ms in the audiovisual paradigm (range: 1047.479–4573.56ms, n = 157; see Fig 3).

In the retrieval part of the test phase, the median RER in the visual paradigm was 0.056 (range: 0–0.86, n = 157), and it was 0.028 (0–0.42, n = 157) in the audiovisual one. The difference was significant (Z = 4.812, p < 0.001; Fig 4). Furthermore, retrieval RTs (RER RTs) were significantly shorter in the audiovisual paradigm (Z = 4.452, p < 0.001 m; Fig 3). The median

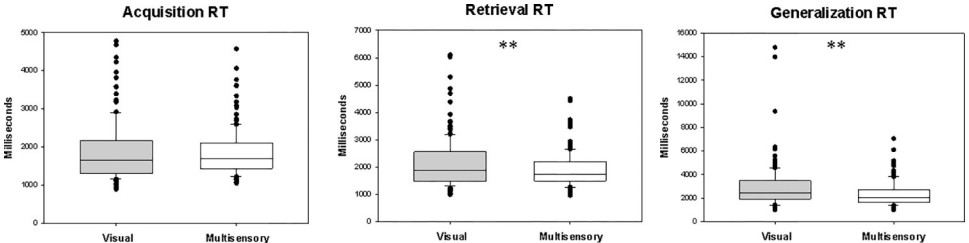

**Fig 3. Reaction times in the visual and multisensory paradigms (in milliseconds).** The reaction times did not differ significantly between the visual and audiovisual paradigms in the acquisition phase. The conventions are the same as in Fig 2.

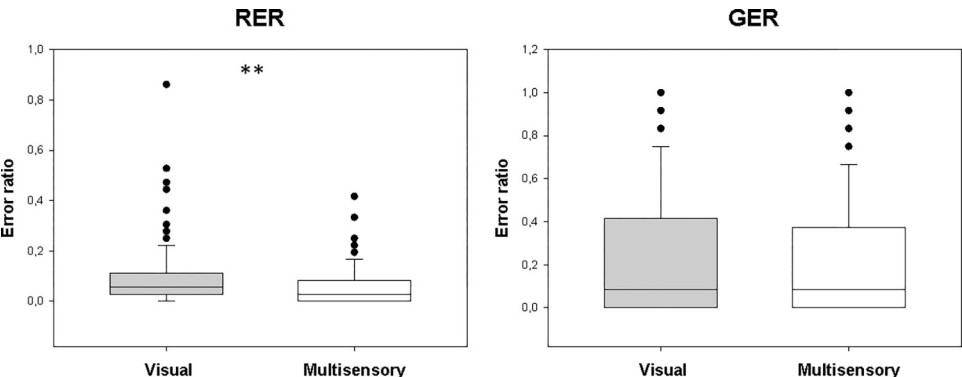

**Fig 4. Performance in the test phase of the visual and audiovisual paradigms.** The test phase can be divided into two parts, retrieval and generalization (see text for details) Performance in these parts is characterized by the retrieval error ratio (RER) and generalization error ratio (GER). RER differed significantly between the paradigms at p<0.01, while GER did not differ significantly between the paradigms. The conventions are the same as in Fig 2.

RT in the visual paradigm was 1869.750 ms (range: 984.625–6103.87 ms, n = 157), and in the audiovisual paradigm it was 1731.171 ms (range: 956.778–4506.25 ms, n = 157; Fig 3).

The median GER in the visual paradigm was 0.083 (range: 0–1.0, n = 157), and it was also 0.083 in the audiovisual paradigm (range: 0–1.0, n = 157). In contrast to NAT, ALER, and RER, GER did not differ significantly between the two paradigms (Z = 1.006, p = 0.315; Fig 4). Generalization RTs (GER RTs), however, were significantly shorter in the audiovisual paradigm (Z = 3.848, p < 0.001). The median generalization RT in the visual test was 2477.917ms (range: 1001.167–14796.50 ms, n = 153). In the audiovisual paradigm, it was 2064.167 (range: 1004.400–7054.000 ms, n = 152; Fig 3).

## The effect of age on performance

Linear regression analysis was performed to analyze the age-dependence of the studied parameters. All the investigated parameters, both in the acquisition and the test phases, showed a significant negative correlation with the age of the participants. That is, performance improved with age in general (see Table 2).

## Performance above and below 12 years of age

Eighty-five of the subjects (54.1%) were younger than 12 years of age, and 72 of them (45.9%) were older than 12 years of age. Descriptive statistics of their performance is shown in Table 3. As for the acquisition phase (as assessed with NAT and ALER), both cohorts' performance was superior in the audiovisual paradigm. This was true for the retrieval part of the test phase as well (RER). However, no such difference was observed in either cohort in the generalization part (GER). The results of the hypothesis tests are given in Table 4.

A comparison of the performance of the two cohorts (below and above 12 years of age) by the studied parameters shows that the older cohort outperformed the younger one in both paradigms and in all parameters. For these comparisons, the Mann-Whitney U test was used. The results are shown in Table 5.

## The correlation of multisensory gain with the age of the children

Correlation analysis between multisensory gain and age revealed significant correlation in the acquisition phase. In most parameters, the gain values were below zero, which means that the

**Table 2. Linear regression results of correlation between age and performance (NAT, ALER, RER, GER, RTs) in both paradigms (V: visual, M: multisensory).**

| Parameter vs age | b* | p |
|---|---|---|
| VNAT | -0,421125 | 0,000000 |
| VALER | -0,473864 | 0,000000 |
| VAcqRT | -0,647056 | 0,000000 |
| VRER | -0,239547 | 0,002514 |
| VRER RT | -0,587382 | 0,000000 |
| VGER | -0,210698 | 0,008079 |
| VGER RT | -0,304428 | 0,000130 |
| MNAT | -0,232480 | 0,003390 |
| MALER | -0,244188 | 0,002056 |
| MAcqRT | -0,511363 | 0,000000 |
| MRER | -0,424617 | 0,000000 |
| MRER RT | -0,492971 | 0,000000 |
| MGER | -0,323081 | 0,000037 |
| MGER RT | -0,401983 | 0,000000 |

multisensory error ratios were frequently lower than the visual ones. Descriptive statistics and correlation coefficients are given in Table 6.

Descriptive statistics of the multisensory gains of the two age groups is shown in Table 7. A comparison of the multisensory gain of the two cohorts (below and above 12 years of age) by the studied parameters shows that the older cohort has smaller gain than the younger, and the differences were significant in the acquisition phase (Table 8).

## Discussion

In this study, we investigated the effectiveness of visual and audiovisual equivalence learning in a large sample of healthy children and adolescents. To our knowledge, we are the first to demonstrate that, in contrast to healthy adults, audiovisual information facilitates equivalence learning in healthy children and adolescents.

Two sensory guided associative learning tests with the same structure were used, one visual [22] and one audiovisual [12]. Both tests were developed in our laboratory, based on the Rutgers Acquired Equivalence Test [1]. The Rutgers Acquired Equivalence Test was originally developed to dissociate the contributions of the basal ganglia and the hippocampi to visual equivalence learning and transfer. Myers and co-workers [1] found that patients with

**Table 3. Descriptive statistics of performance (NAT, ALER, RER, GER) in both paradigms (V: visual, M: multisensory) below and above 12 years of age.**

| Parameter | < 12 years | | | | >12 years | | | |
|---|---|---|---|---|---|---|---|---|
| | N | Median | Minimum | Maximum | N | Median | Minimum | Maximum |
| VNAT | 85 | 73.000 | 42.00 | 204.00 | 72 | 56.500 | 41.00 | 136.00 |
| VALER | 85 | 0.104 | 0.00 | 0.34 | 72 | 0.060 | 0.00 | 0.19 |
| VRER | 85 | 0.083 | 0.00 | 0.53 | 72 | 0.028 | 0.00 | 0.86 |
| VGER | 85 | 0.250 | 0.00 | 1.00 | 72 | 0.083 | 0.00 | 1.00 |
| MNAT | 85 | 55.000 | 41.00 | 134.00 | 72 | 50.500 | 41.00 | 123.00 |
| MALER | 85 | 0.056 | 0.00 | 0.36 | 72 | 0.042 | 0.00 | 0.29 |
| MRER | 85 | 0.056 | 0.00 | 0.42 | 72 | 0.000 | 0.00 | 0.17 |
| MGER | 85 | 0.167 | 0.00 | 1.00 | 72 | 0.000 | 0.00 | 1.00 |

**Table 4. Between-paradigm comparisons below and above 12 years of age.** Results of the hypothesis tests. The conventions are the same as in Table 3.

| Comparison | < 12 years | | | >12 years | | |
|---|---|---|---|---|---|---|
| | N | Z | p | N | Z | p |
| **VNAT vs. MNAT** | 85 | 4.816559 | 0.000001 | 72 | 2.111445 | 0.034735 |
| **VALER vs. MALER** | 85 | 4.526674 | 0.000006 | 72 | 1.787709 | 0.073824 |
| **VRER vs. MRER** | 85 | 3.248503 | 0.001160 | 72 | 4.098464 | 0.000042 |
| **VGER vs. MGER** | 85 | 0.193603 | 0.846487 | 72 | 1.661560 | 0.096602 |

Parkinson's disease exhibited poor performance when forming the visual associations, while patients with hippocampal atrophy were characterized by poor transfer. In this way, the authors demonstrated that the basal ganglia and the hippocampi are key structures in associative equivalence acquisition and the transfer of the equivalence rule to new stimuli, respectively, and that the test is capable of picking up suboptimal function of these structures. Since then, it has become widely recognized in the literature the basal ganglia have a key role in the association of stimuli [23, 24], while transfer is linked mainly the hippocampi/medial temporal lobe [3, 25]. The Rutgers paradigm has been applied to learn about associative learning/equivalence learning in various psychiatric and neurological disorders characterized by the dysfunction of the basal ganglia and the hippocampi [22, 26–29] and also in healthy subjects [30, 31].

Since the key brain structures involved in sensory guided associative/equivalence learning (the basal ganglia and the hippocampi) process not only visual but also auditory and combined audiovisual information [4–7], we have developed a new multisensory (audiovisual) version of the Rutgers Acquired Equivalence Test to enable the exploration of multisensory guided associative/equivalence learning. We first used this new test to explore this kind of learning in healthy adults [12]. We also compared the results with those obtained with the original visual-only paradigm. The results revealed that performance throughout the test was fairly independent of stimulus modality [12]. The same was true for reaction times. We concluded that the effectiveness of sensory guided associative/equivalence learning does not depend on the modality of the applied stimuli in healthy adults.

The findings presented in this study show a different picture. In terms of performance (assessed as error ratios in the various parts of the test) children and adolescents seem to benefit significantly from multimodality in acquisition and retrieval, but not in generalization. Reaction times, however, were significantly shorter in the audiovisual paradigm, even in the generalization part of the test phase. In other words, in the audiovisual paradigm, the subjects performed at approximately the same level as in the visual paradigm, but with significantly shorter reaction times. This all suggests that healthy children and adolescents learn and

**Table 5. Parameter-by-parameter comparison of performance between the two cohorts (below and above 12 years of age).** Results of the hypothesis tests (Mann-Whitney U). The conventions are the same as in Table 3.

| Parameter | Z | p | N below 12 | N above 12 |
|---|---|---|---|---|
| **VNAT** | 4.370017 | 0.000012 | 85 | 72 |
| **VALER** | 4.850878 | 0.000001 | 85 | 72 |
| **VRER** | 3.748245 | 0.000178 | 85 | 72 |
| **VGER** | 2.680841 | 0.007344 | 85 | 72 |
| **MNAT** | 1.794860 | 0.072677 | 85 | 72 |
| **MALER** | 1.895259 | 0.058059 | 85 | 72 |
| **MRER** | 4.845593 | 0.000001 | 85 | 72 |
| **MGER** | 4.100524 | 0.000041 | 85 | 72 |

**Table 6. Descriptive statistics of multisensory gain and correlation coefficients (Spearman's ρ).**

| Parameter | Median | Minimum | Maximum | ρ |
|---|---|---|---|---|
| **GAIN NAT** | -8,000 | -140,0 | 64,000 | 0,246756 |
| **GAIN ALER** | -0,022 | -0,2 | 0,248 | 0,255976 |
| **GAIN RER** | -0,028 | -0,8 | 0,333 | 0,029085 |
| **GAIN GER** | 0,000 | -1,0 | 1,000 | -0,054252 |

Significant correlations (p < 0.05) are marked in light gray. The conventions are the same as in Figs 3 and 4.

retrieve associations more efficiently if the stimuli are of different modalities. Generalization does not seem to be facilitated by multimodality in terms of performance, but the significantly shorter reaction times suggest that a certain level of facilitation is present also in this part of the paradigm.

Multisensory integration plays an important role not only in sensory-motor but also in cognitive functions. Bimodal (or multimodal) facilitation could enhance sensory perception [32], object recognition [33, 34], emotional change recognition [35], face and voice recognition [36], and person recognition [37]. Semantic congruence can strengthen multisensory integration [38], but in the case of our stimuli, such a congruency is negligible if it exists at all. Thus, it is safe to assume that in this study multisensory integration facilitated performance without semantic interference. Multisensory integration has been described at various levels of observation. It has been described in detail at the single-cell level [39–42] in both the neocortex [8] and in subcortical structures [5, 9, 43]. It is also well documented in various cognitive functions at the behavioral level [10, 11, 44]. Multisensory integration has been shown to influence various cognitive-behavioral parameters such as reaction time, accuracy of answers, or perception thresholds [45–48]. Our results suggest that multisensory integration enhances the learning and retrieval of associations in healthy children and adolescents, and in this sense our results are in agreement with the literature.

The reason for the superiority of audiovisual information as input for equivalence learning in children and adolescents but not in adults [12] can be that visually guided equivalence learning is still in development in childhood and adolescence [30], that is, it has not yet reached its optimum. It can be hypothesized that the additional modality enhances the suboptimal performance that is observed in the unimodal paradigm. By adulthood, however, visual equivalence learning reaches its optimum, there is no significant development anymore [12], so the beneficial effect of multimodality disappears.

The developmental patterns of multisensory integration depend on the applied modalities and cognitive tasks. For instance, the integration auditory and tactile modalities goes through the most significant development between 8 and 10 years of age, while for the auditory and visual modalities, this falls between 11 and 12 years of age [14–19]. Incidental category learning is an intriguing exception, as children as young as 6 years of age use audiovisual stimuli

**Table 7. Descriptive statistics of multisensory gain in all parameters below and above 12 years of age.** The conventions are the same as in Figs 3 and 4.

| Parameter | < 12 years | | | | >12 years | | | |
|---|---|---|---|---|---|---|---|---|
| | N | Median | Minimum | Maximum | N | Median | Minimum | Maximum |
| **GAIN NAT** | 85 | -16,000 | -140,0 | 63,000 | 72 | -4,000 | -85,00 | 64,000 |
| **GAIN ALER** | 85 | -0,045 | -0,2 | 0,231 | 72 | -0,017 | -0,15 | 0,248 |
| **GAIN RER** | 85 | -0,028 | -0,5 | 0,333 | 72 | -0,028 | -0,83 | 0,139 |
| **GAIN GER** | 85 | 0,000 | -0,8 | 1,000 | 72 | 0,000 | -1,00 | 1,000 |

**Table 8. Comparison of multisensory gains between the two cohorts (below and above 12 years of age).** Results of the hypothesis tests (Mann-Whitney U). The conventions are the same as in Figs 3 and 4.

| GAIN | Z | p |
|------|------|------|
| NAT | -2,73368261 | 0,006253 |
| ALER | -2,63680597 | 0,008369 |
| RER | -0,70808016 | 0,474929 |
| GER | -1,52008253 | 0,125694 |

efficiently for this cognitive task [49, 50]. In our study, subjects both below and above 12 years of age integrated auditory and visual signals successfully in an equivalence learning task, and the performance of both cohorts was superior in the audiovisual test as compared to the visual test. At the same time, we observed a significant performance improvement: when subjects below and above 12 years of age were compared, subjects above 12 years of age significantly outperformed subjects below 12 years of age in all parameters and in both test paradigms.

Our results demonstrate that multisensory stimuli significantly enhance association learning and retrieval in the context of sensory guided equivalence learning in healthy children and adolescents. Furthermore, our results suggest that audiovisually guided equivalence learning are still in development in childhood and adolescence, which is especially well illustrated by the difference in audiovisual gain between subjects below and above 12 years of age.

## Supporting information

**S1 Data.**
(XLSX)

## Acknowledgments

The authors thank András Puszta and Ákos Pertich for their help with data collection.

## Author Contributions

**Conceptualization:** Gabriella Eördegh, Szabolcs Kéri, Gábor Braunitzer, Attila Nagy.

**Data curation:** Kálmán Tót, Ádám Kiss.

**Formal analysis:** Gabriella Eördegh.

**Funding acquisition:** Attila Nagy.

**Investigation:** Kálmán Tót, Ádám Kiss.

**Methodology:** Gabriella Eördegh, Kálmán Tót, Szabolcs Kéri, Attila Nagy.

**Project administration:** Attila Nagy.

**Resources:** Attila Nagy.

**Supervision:** Gabriella Eördegh, Szabolcs Kéri, Gábor Braunitzer, Attila Nagy.

**Validation:** Kálmán Tót.

**Visualization:** Kálmán Tót, Ádám Kiss.

**Writing – original draft:** Gabriella Eördegh, Kálmán Tót, Attila Nagy.

**Writing – review & editing:** Szabolcs Kéri, Gábor Braunitzer.

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
