## [Decision Letter · Decision Letter 0]

25 Feb 2022

PONE-D-21-24459Multisensory Stimuli Enhance the Effectiveness of Equivalence Learning in Healthy Children and AdolescentsPLOS ONE

Dear Dr. Eördegh,

Thank you for submitting your manuscript to PLOS ONE. After careful consideration, we feel that it has merit but does not fully meet PLOS ONE’s publication criteria as it currently stands. Therefore, we invite you to submit a revised version of the manuscript that addresses the points raised during the review process.

Please address the revisions recommended by both reviewers.

We look forward to receiving your revised manuscript.

Kind regards,

Bernadette Ann Murphy, PhD

Academic Editor

PLOS ONE

Journal Requirements:

2. Please note that according to our submission guidelines (http://journals.plos.org/plosone/s/submission-guidelines), outmoded terms and potentially stigmatizing labels should be changed to more current, acceptable terminology. To this effect, please use "White or "of western European descent" instead of "Caucasian".

"The authors thank András Puszta and Ákos Pertich for their help with data collection. This work was supported by a grant from SZTE ÁOK-KKA Grant No.:2019/270-62-2. KT was supported by EFOP 3.6.3-VEKOP-16-2017-00009 grant."

"AN: SZTE ÁOK-KKA Grant No.:2019/270-62-2

Funder: University of Szeged, Faculty of Medicine, Szeged, Hungary

http://www.med.u-szeged.hu/karunkrol/kari-palyazatok/aok-kari-kutatasi-alap-181005

KT: EFOP 3.6.3-VEKOP-16-2017-00009

Funder: Goverment of Hungary

https://www.palyazat.gov.hu/efop-363-vekop-16-felsoktatsi-hallgatk-tudomnyos-mhelyeinek-s-programjainak-tmogatsa#

Reviewers' comments:

Reviewer's Responses to Questions

**Comments to the Author**

1. Is the manuscript technically sound, and do the data support the conclusions?

Reviewer #1: Yes

Reviewer #2: Partly

2. Has the statistical analysis been performed appropriately and rigorously? 

Reviewer #1: Yes

Reviewer #2: Yes

3. Have the authors made all data underlying the findings in their manuscript fully available?

Reviewer #1: Yes

Reviewer #2: No

4. Is the manuscript presented in an intelligible fashion and written in standard English?

Reviewer #1: Yes

Reviewer #2: Yes

5. Review Comments to the Author

Reviewer #1: Thank you for the opportunity to review this article. The topic and work are interesting and provide a novel contribution to the field of learning and sensory processing/multisensory integration. Overall, the article was well written, providing clear details on the methodology used and results yielded. There are a number of sections where further detail could be provided for clarity.

Introduction: The introduction was concise. Although, several lines seem to be missing references, such as those referring to the neurological substrates involved (i.e. lines 50, 55/56, etc.).

Methods:

• Line 84 – how were participants defined as healthy? Were any screening questionnaires etc. administered?

• Line 128 – when you say “it was slightly modified”, how so? Does this refer to translating it to Hungarian? Could more detail be provided or clarity of statement be improved?

• Paragraph starting at line 153 – were the auditory and visual stimuli presented simultaneously? Or, were they presented at different times (i.e. antecedent and consequent)? The timing of the conditions would likely have an effect on how the stimuli are processed, and therefore integrated or not. Line 159 states the faces were presented simultaneously, was this at the same time as the sound as well?

• Line 159 – audiovisual condition is lacking semantic congruence. Could this affect the results found? As literature, such as Laurienti et al., 2004, has shown that multisensory integration is strengthened when stimuli are semantically congruent. I.e. the auditory cues here are words/sounds that are incongruent with the visual cues.

• Line 178 – Reaction Time defined as (RT)? Sometimes use RT but other times use reaction time, check for consistency.

• Line 180 – “RT values over 3SD were excluded”. 3SD over what? Each participant’s individual average RT, the group average, etc.? I suspect the former, but this isn’t clear.

• Was handedness checked or confirmed? If so, was this proportionally matched between groups? What hand did they respond with, or did they use both hands? Did all participants have to respond with the same hand, or did they use their dominant hand? Line 100 - mention how responses were given (M and X keys), but not many details provided, please elaborate.

Results:

• Line 237 – “leaning” should be “learning”.

• Line 211 – sentence structure - where the “0.051” is placed seems strange, was this supposed to be before it reads “in the multisensory paradigm”?

Discussion:

• Line 308 paragraph – could this be a result of either a) stimulus incompatibility/semantic incongruence or b) the timing of the stimulus, possibly reducing the likelihood of them being integrated as a multisensory condition?

• Line 328 – Interesting, what are some potential implications for this? As previously stated that MSI wasn’t present in children (10-11 years old; line 79). Would be interesting for this to be elaborated on, as it was a main finding.

• Line 289 - discuss that audiovisual information facilitated equivalence learning in children. There were differences between groups (<12 and >12) but in the intro, stated that MSI didn’t occur until 10-11 years old. What do you think resulted in these differences if not the integration of the stimuli? Can this be elaborated on?

• Paragraph from line 332 to 343 – seems out of place with respect to the preceding paragraph. Would be helpful to weave in how this information pertains to this particular study and results itself.

• Line 359 – improvement in performance, using what metric? RT, accuracy, both, etc.?

Reviewer #2: The study compared a visual and a crossmodal version of an association learning test (Rutgers Acquired Equivalence Test) in an impressive sample of 157 children and adolescents. It was found that RT was consistently faster and that performance (error rate) during acquisition and during retrieval of previously learned associations was better in the crossmodal as compared to the unimodal condition, whereas performance did not differ for generalization to new but predictable associations.

Overall, this is a solid and technically sound contribution. My comments mainly pertain to the conceptualization of multisensory processing and the somewhat arbitrary distinction between children younger and older than 12 years of age:

l. 28 "Performance during the acquisition phase (building of associations), which primarily depends on the function of the basal ganglia": the present study does not assess neural mechanisms, I would thus recommend to attenuate this statement here and in other parts of the manuscript, e.g. "which has been suggested to primarily depend…".

l. 62 and elsewhere "Multisensory information could mean more than the sum of different modalities": The authors seem to allude to findings of superadditivity in neural processing of crossmodal stimuli. However, the wording is unfortunate and it is not clear what superadditivity would mean in the context of the present (behavioral) task?

l. 77ff. "Multisensory integration is not detectable until the ages of 8–10 years in auditory and tactile modalities [13, 14], and audiovisual integration does not appear until the ages of 11–12 years [15-18]": I think that the available literature is in stark contrast to this statement. While the cited studies may show that integration is not necessarily optimal in younger children, it seems outright wrong to assume that this indicates that children up to 12 years do not integrate audiovisual stimuli at all. For examples of adult-like audiovisual integration in accord with Bayesian causal inference principles in children as young as 5 years old, see e.g. recent work by Rohlf and colleagues (Rohlf et al., 2020, Curr Biol; Rohlf et al., 2021, Multisens Res).

Based on these findings, the presumed boundary at age 12 seems arbitrary. Why not test a correlation with age instead? For example, scatterplots of age versus the difference between visual and crossmodal conditions could be provided for each performance measure.

Based on the equivalence of the present findings in children below and above 12 years, the authors conclude that children and adolescents <18 years benefit from crossmodal condition but adults >18 years do not. This seems arbitrary as well, why should 18 years of age be a cut-off age for the crossmodal benefit in association learning?

Moreover, the type of stimuli used here seem to test a different type of multisensory processing (association learning) than the integration of redundant stimulus features (e.g., spatial or temporal integration) and likely depend on different neural mechanisms. This should be reflected in the discussion of the multisensory literature.

l. 178ff. "Reaction times from the appearance of the stimuli until the participant’s decision were also registered for each answer with millisecond accuracy": RTs were measured with (different) standard computer keyboards which almost certainly introduced delays and jitter to the measured RTs, thus "millisecond accuracy" was likely not achieved.

l. 285 "the effect of multisensory audiovisual stimuli in contrast to clear visual ones": What is meant with "clear" visual stimuli, this sounds as if the audiovisual stimuli were not "clear"?

l. 332ff. lists textbook knowledge about multisensory processing in a very vague form, this paragraph should be revised.

l. 344ff. suggests a ceiling effect in the employed task in adults. Thus, it seems unsurprising if no multisensory enhancement was observed in the previous adult study.

Visual face stimuli were used as antecedent stimuli in the visual condition but as consequent stimuli in the crossmodal condition. Thus, stimuli differed not only in terms of sensory modality but also in terms of stimulus type between the two conditions. It seems that a better matching crossmodal condition would have been to use voice recordings of male/female adult/child speakers as antecedents and the same fish stimuli as consequents. I wonder whether the change in stimulus types might have had any effects on task difficulty that are independent of whether the stimuli were presented visually or crossmodally (and thus could explain the findings without assuming a multisensory benefit)?

Data availability: Authors indicated that "all relevant data will be within the manuscript and its Supporting Information files". However, they were not in the reviewer's version of the manuscript and thus I was unable to assess this point.

6. PLOS authors have the option to publish the peer review history of their article (what does this mean?). If published, this will include your full peer review and any attached files.

Reviewer #1: No

Reviewer #2: No

---

## [Author Response · Author response to Decision Letter 0]

19 Apr 2022

First of all, we would like to express our gratitude for the scholarly and highly helpful criticism of the Editorial Board and both Referees, which helped us to improve the quality of our study. We discussed all criticisms and suggestions of the Referees and made changes accordingly. The suggestions are answered below point-by-point; textual changes are marked in the final manuscript.

Reviewer #1: Thank you for the opportunity to review this article. The topic and work are interesting and provide a novel contribution to the field of learning and sensory processing/multisensory integration. Overall, the article was well written, providing clear details on the methodology used and results yielded. There are a number of sections where further detail could be provided for clarity.

1.Introduction: The introduction was concise. Although, several lines seem to be missing references, such as those referring to the neurological substrates involved (i.e. lines 50, 55/56, etc.).

Answer: We have added the missing references.

Methods:

2. Line 84 – how were participants defined as healthy? Were any screening questionnaires etc. administered?

Answer: We made clearer this paragraph in the methods: “The participants were recruited on a voluntary basis, received no compensation for their participation, and they were free to quit at any time without any consequence. The volunteers and their parents were informed about the aims and procedures of the study, and their medical history was taken with emphasis on neurological, ontological, psychiatric or chronic somatic disorders. Volunteers with such disorders in their history were not eligible for the study. Any regularly taken medication was recorded. The volunteers were also tested with the Ishihara plates to exclude color blindness. As all volunteers were under 18 years of age, the informed consent form was signed by their parents for them as required by the law. All volunteers were of Caucasian descent and they were all native speakers of the Hungarian language. The study protocol followed the tenets of the Declaration of Helsinki in all respects, and it was approved by the Ministry of Human Resources (11818-6/2017/EÜIG).”

3. Line 128 – when you say “it was slightly modified”, how so? Does this refer to translating it to Hungarian? Could more detail be provided or clarity of statement be improved?

Answer: We rephrased the criticized paragraph: “The basis of the applied visual associative test was the Rutgers Acquired Equivalence Test [1]. It was rewritten in Assembly for Windows, translated to Hungarian, and slightly modified (more trials in test phases to get more accurate information about the hippocampal functions) [22] with the written permission of professor Catherine E. Myers (Rutgers University), head of the research group where the test paradigm was originally developed.”

4. Paragraph starting at line 153 – were the auditory and visual stimuli presented simultaneously? Or, were they presented at different times (i.e. antecedent and consequent)? The timing of the conditions would likely have an effect on how the stimuli are processed, and therefore integrated or not. Line 159 states the faces were presented simultaneously, was this at the same time as the sound as well?

Answer: We added to the manuscript: “In each trial, the subjects simultaneously heard a sound (SPL=60 dB) through a loudspeaker and saw two faces on the right and left sides of the screen.”

5. Line 159 – audiovisual condition is lacking semantic congruence. Could this affect the results found? As literature, such as Laurienti et al., 2004, has shown that multisensory integration is strengthened when stimuli are semantically congruent. I.e. the auditory cues here are words/sounds that are incongruent with the visual cues.

Answer: We added to the Discussion: “Semantic congruence can strengthen multisensory integration (Semantic congruence is a critical factor in multisensory behavioral performance. Laurienti PJ, Kraft RA, Maldjian JA, Burdette JH, Wallace MT.Exp Brain Res. 2004 Oct;158(4):405-14. doi: 10.1007/s00221-004-1913-2. Epub 2004 Jun 18.PMID: 15221173), but in the case of our stimuli, such a congruency is negligible if it exists at all. Thus, it is safe to assume that in this study multisensory integration facilitated performance without semantic interference. Multisensory integration has been described at various levels of observation.”

6. Line 178 – Reaction Time defined as (RT)? Sometimes use RT but other times use reaction time, check for consistency.

Answer: We introduced the RT abbreviation at the first appearance of the Reaction Time.

7. Line 180 – “RT values over 3SD were excluded”. 3SD over what? Each participant’s individual average RT, the group average, etc.? I suspect the former, but this isn’t clear.

Answer: We rephrased the sentence: RT values over 3 SD of each participant’s individual average RT were excluded from further analysis.

8. Was handedness checked or confirmed? If so, was this proportionally matched between groups? What hand did they respond with, or did they use both hands? Did all participants have to respond with the same hand, or did they use their dominant hand? Line 100 - mention how responses were given (M and X keys), but not many details provided, please elaborate.

Answer: We specified this question: “The keys X and M were labeled as “left” and “right” on the laptop’s keyboard. The subjects used these keys to indicate their choices in both test paradigms. The participants used both hands for the responses.”

Results:

9. Line 237 – “leaning” should be “learning”.

Answer: Corrected

10. Line 211 – sentence structure - where the “0.051” is placed seems strange, was this supposed to be before it reads “in the multisensory paradigm”?

Answer: Done

Discussion:

11. Line 308 paragraph – could this be a result of either a) stimulus incompatibility/semantic incongruence or b) the timing of the stimulus, possibly reducing the likelihood of them being integrated as a multisensory condition?

Answer: We added to the Discussion: “Semantic congruence can strengthen multisensory integration (Semantic congruence is a critical factor in multisensory behavioral performance. Laurienti PJ, Kraft RA, Maldjian JA, Burdette JH, Wallace MT.Exp Brain Res. 2004 Oct;158(4):405-14. doi: 10.1007/s00221-004-1913-2. Epub 2004 Jun 18.PMID: 15221173), but in the case of our stimuli, such a congruency is negligible if it exists at all. Thus, it is safe to assume that in this study multisensory integration facilitated performance without semantic interference. Multisensory integration has been described at various levels of observation.”

In each trial, the auditory and the visual stimuli appeared simultaneously. Thus the participant heard simultaneously a sound (SPL=60 dB) through a loudspeaker and saw two faces on the right and left sides of the screen. Thus, the timing of the stimuli were constant and consistent and it could not influence the results.

12. Line 328 – Interesting, what are some potential implications for this? As previously stated that MSI wasn’t present in children (10-11 years old; line 79). Would be interesting for this to be elaborated on, as it was a main finding.

Answer: We gave the answer to this question in answer to line 289.

13. Line 289 - discuss that audiovisual information facilitated equivalence learning in children. There were differences between groups (<12 and >12) but in the intro, stated that MSI didn’t occur until 10-11 years old. What do you think resulted in these differences if not the integration of the stimuli? Can this be elaborated on?

Answer: We have rephrased the discussion: “The reason for the superiority of audiovisual information as input for equivalence learning in children and adolescents but not in adults [12] can be that visually guided equivalence learning is still in development in childhood and adolescence [30], that is, it has not yet reached its optimum. It can be hypothesized that the additional modality enhances the suboptimal performance that is observed in the unimodal paradigm. By adulthood, however, visual equivalence learning reaches its optimum, there is no significant development anymore [12], so the beneficial effect of multimodality disappears.

The developmental patterns of multisensory integration depend on the applied modalities and cognitive tasks. For instance, the integration auditory and tactile modalities goes through the most significant development between 8 and 10 years of age, while for the auditory and visual modalities, this falls between 11 and 12 years of age [14-19]. Incidental category learning is an intriguing exception, as children as young as 6 years of age use audiovisual stimuli efficiently for this cognitive task [49, 50]. In our study, subjects both below and above 12 years of age integrated auditory and visual signals successfully in an equivalence learning task, and the performance of both cohorts was superior in the audiovisual test as compared to the visual test. At the same time, we observed a significant performance improvement: when subjects below and above 12 years of age were compared, subjects above 12 years of age significantly outperformed subjects below 12 years of age in all parameters and in both test paradigms.”

14. Line 359 – improvement in performance, using what metric? RT, accuracy, both, etc.?

Answer: We amended the sentence as follow: “At the same time, we observed a significant performance improvement: when subjects below and above 12 years of age were compared, subjects above 12 years of age significantly outperformed subjects below 12 years of age in all parameters and in both test paradigms.”

Reviewer #2: The study compared a visual and a crossmodal version of an association learning test (Rutgers Acquired Equivalence Test) in an impressive sample of 157 children and adolescents. It was found that RT was consistently faster and that performance (error rate) during acquisition and during retrieval of previously learned associations was better in the crossmodal as compared to the unimodal condition, whereas performance did not differ for generalization to new but predictable associations.

Overall, this is a solid and technically sound contribution. My comments mainly pertain to the conceptualization of multisensory processing and the somewhat arbitrary distinction between children younger and older than 12 years of age:

l. 28 "Performance during the acquisition phase (building of associations), which primarily depends on the function of the basal ganglia": the present study does not assess neural mechanisms, I would thus recommend to attenuate this statement here and in other parts of the manuscript, e.g. "which has been suggested to primarily depend…".

Answer: We have deleted from the Abstract the speculation about the role of neuronal structures in this learning process, and added the missing references to the Introduction section of the manuscript.

2. 62 and elsewhere "Multisensory information could mean more than the sum of different modalities": The authors seem to allude to findings of superadditivity in neural processing of crossmodal stimuli. However, the wording is unfortunate and it is not clear what superadditivity would mean in the context of the present (behavioral) task?

Answer: Superadditivity could mean that the performances in the multisensory paradigm is better that those in the two unimodal visual or auditory paradigms. Because of the absence of a clear auditory associative learning test we were unable to check this statement. Thus we have deleted the speculations about the superadditivity in the learning processes.

3. 77ff. "Multisensory integration is not detectable until the ages of 8–10 years in auditory and tactile modalities [13, 14], and audiovisual integration does not appear until the ages of 11–12 years [15-18]": I think that the available literature is in stark contrast to this statement. While the cited studies may show that integration is not necessarily optimal in younger children, it seems outright wrong to assume that this indicates that children up to 12 years do not integrate audiovisual stimuli at all. For examples of adult-like audiovisual integration in accord with Bayesian causal inference principles in children as young as 5 years old, see e.g. recent work by Rohlf and colleagues (Rohlf et al., 2020, Curr Biol; Rohlf et al., 2021, Multisens Res).

Answer: We rephrased the introduction and we inserted the two recommended references. “Concerning the development of multisensory integration in childhood, the available data are controversial and they strongly depend on stimulus modality and the studies cognitive function. The literature distinguishes between two main types of multisensory integration: the integration of different modalities and the integration of redundant stimulus features (e.g., spatial or temporal integration). The integration of different modalities is not detectable until 8 to 10 years of age in the auditory and tactile modalities [14, 15], and audiovisual integration is suboptimal (but detectable) until 11 to 12 years of age [16-21]. Therefore, in this study we also sought to investigate if there was a difference in participants’ performance depending on whether they were above or below 12 years of age.”

4. Based on these findings, the presumed boundary at age 12 seems arbitrary. Why not test a correlation with age instead? For example, scatterplots of age versus the difference between visual and crossmodal conditions could be provided for each performance measure.

Answer: We wrote a new paragraph in the Results section with the title: “The effect of age on performance”, and showed the asked correlations: “Linear regression analysis was performed to analyze the age-dependence of the studied parameters. All the investigated parameters, both in the acquisition and the test phases, showed a significant negative correlation with the age of the participants. That is, performance improved with age in general (see Table 2).”

5. Based on the equivalence of the present findings in children below and above 12 years, the authors conclude that children and adolescents <18 years benefit from crossmodal condition but adults >18 years do not. This seems arbitrary as well, why should 18 years of age be a cut-off age for the crossmodal benefit in association learning?

Answer: This is a technical consideration only. In our country (Hungary) this the border between the adolescents and adults.

Furthermore, the results of the adult and children populations came from different studies. We stated in our previous study (Eördegh et al., 2019) that the multisensory stimuli cannot enhance the performances if adults (over 18 years old). But this is not the case in children and adolescents under 18 years old where the multisensory stimuli enhance the performances. 

However, the investigation of the correlation between the age and the performances in the entire data set (under and above 18 years old together) is a topic of a new analysis and a new manuscript.

6. Moreover, the type of stimuli used here seem to test a different type of multisensory processing (association learning) than the integration of redundant stimulus features (e.g., spatial or temporal integration) and likely depend on different neural mechanisms. This should be reflected in the discussion of the multisensory literature.

Answer: The multisensory integration is a broad concept, and is sometimes used in the literature not with perfect accuracy. We amended the paragraph with the sentence: “The literature distinguishes between two main types of multisensory integration: the integration of different modalities and the integration of redundant stimulus features (e.g., spatial or temporal integration).”

We added to the Discussion: “Multisensory integration has been shown to influence various cognitive-behavioral parameters such as reaction time, accuracy of answers, or perception thresholds [45-48]. Our results suggest that multisensory integration enhances the learning and retrieval of associations in healthy children and adolescents, and in this sense our results are in agreement with the literature.”

7. l. 178ff. "Reaction times from the appearance of the stimuli until the participant’s decision were also registered for each answer with millisecond accuracy": RTs were measured with (different) standard computer keyboards which almost certainly introduced delays and jitter to the measured RTs, thus "millisecond accuracy" was likely not achieved.

Answer: The Reviewer has right there is no sense to use the ms accuracy. Thank you for this suggestion. We deleted the "millisecond accuracy" from the Materials and methods section.

8. 285 "the effect of multisensory audiovisual stimuli in contrast to clear visual ones": What is meant with "clear" visual stimuli, this sounds as if the audiovisual stimuli were not "clear"?

Answer: We changed the word „clear” to „single”.

9. l. 332ff. lists textbook knowledge about multisensory processing in a very vague form, this paragraph should be revised.

Answer: We have rephrased the criticized paragraph: “Multisensory integration has been described at various levels of observation. It has been described in detail at the single-cell level [39-42] in both the neocortex [8] and in subcortical structures [5, 9, 43]. It is also well documented in various cognitive functions at the behavioral level [10, 11, 44].”

10. l. 344ff. suggests a ceiling effect in the employed task in adults. Thus, it seems unsurprising if no multisensory enhancement was observed in the previous adult study.

Answer: The results of the adult and children populations came from different studies. We stated in our previous study (Eördegh et al., 2019) that the multisensory stimuli can not enhance the performances if adults (over 18 years old). But this is not the case in children and adolescents under 18 years old where the multisensory stimuli enhance the performances. 

However, the investigation of the correlation between the age and the performances in the entire data set (under and above 18 years old together) is a topic of a new analysis and a new manuscript.

11. Visual face stimuli were used as antecedent stimuli in the visual condition but as consequent stimuli in the crossmodal condition. Thus, stimuli differed not only in terms of sensory modality but also in terms of stimulus type between the two conditions. It seems that a better matching crossmodal condition would have been to use voice recordings of male/female adult/child speakers as antecedents and the same fish stimuli as consequents. I wonder whether the change in stimulus types might have had any effects on task difficulty that are independent of whether the stimuli were presented visually or crossmodally (and thus could explain the findings without assuming a multisensory benefit)?

Answer: The multisensory paradigm, which is our development and published earlier (see Eördegh et al., 2019), has this arrangement (faces as consequent stimuli in the crossmodal condition). This is a published and already validated test in a huge healthy human population and in patients with obsessive compulsive disorder (Pertich et al., 2019). However, it is an interesting question whether other audiovisual stimulus combinations could influence the effectiveness of the associative learning. We argue that the application of the fishes instead of the faces could not influence significantly the performances.

12. Data availability: Authors indicated that "all relevant data will be within the manuscript and its Supporting Information files". However, they were not in the reviewer's version of the manuscript and thus I was unable to assess this point.

Answer: The statement was: "All relevant data will be within the manuscript and its Supporting Information files, after the acceptance of the manuscript." In this new submission we have provided all of the data (NAT, ALER, RER, GER, RT, ages) in an attached excel file.

---

## [Decision Letter · Decision Letter 1]

11 May 2022

PONE-D-21-24459R1

Multisensory Stimuli Enhance the Effectiveness of Equivalence Learning in Healthy Children and Adolescents

PLOS ONE

Dear Dr. Eördegh,

Thank you for submitting your manuscript to PLOS ONE. After careful consideration, we feel that it has merit but does not fully meet PLOS ONE’s publication criteria as it currently stands. Therefore, we invite you to submit a revised version of the manuscript that addresses the points raised during the review process.

We look forward to receiving your revised manuscript.

Kind regards,

Vanessa Carels

Staff Editor

PLOS ONE

Reviewers' comments:

Reviewer's Responses to Questions

**Comments to the Author**

1. If the authors have adequately addressed your comments raised in a previous round of review and you feel that this manuscript is now acceptable for publication, you may indicate that here to bypass the “Comments to the Author” section, enter your conflict of interest statement in the “Confidential to Editor” section, and submit your "Accept" recommendation.

Reviewer #1: All comments have been addressed

Reviewer #2: (No Response)

2. Is the manuscript technically sound, and do the data support the conclusions?

Reviewer #1: Yes

Reviewer #2: Partly

3. Has the statistical analysis been performed appropriately and rigorously? 

Reviewer #1: Yes

Reviewer #2: Yes

4. Have the authors made all data underlying the findings in their manuscript fully available?

Reviewer #1: Yes

Reviewer #2: Yes

5. Is the manuscript presented in an intelligible fashion and written in standard English?

Reviewer #1: Yes

Reviewer #2: Yes

6. Review Comments to the Author

Reviewer #1: The authors have adequately addressed all review comments. This work makes an interesting contribution to the field of equivalence learning and the role of sensory modalities (visual and multisensory) in various age groups.

A few minor English/grammar corrections to note:

Refer to lines 22, 25, 411, and 525 in the tracked version

Reviewer #2: The authors have largely addressed the points raised in the reviews, and the manuscript has been improved. However, I have one remaining comment:

Authors now report correlations between age and performance in each performance metric individually (i.e., separately for the V and the AV paradigms), and find, maybe not surprisingly, that performance generally increases with age. However, my original comment pertained to the correlation between age and the size of the multisensory gain (i.e., the performance difference between V and AV paradigms, e.g., MNAT minus VNAT). At least for NAT and ALER, the results presented in Tables 3 and 4 seem to suggest that the multisensory gain becomes smaller in >12 as compared to <12, and I wonder if there would be a significant correlation here? If yes, this would indicate that the multisensory gain actually decreases and becomes more adult-like over development, which would be different to the current interpretation that there is no difference until age 18.

Similarly, in Table 5 the authors did not directly test whether there is a significant difference between <12 and >12 in the multisensory gain, they only compare V and AV metrics separately between age groups. I think it would be crucial to directly compare the size of the multisensory gain between age groups (note that multisensory gain could be significant in both age groups as shown in Table 4 but still be significantly smaller in >12 than in <12).

7. PLOS authors have the option to publish the peer review history of their article (what does this mean?). If published, this will include your full peer review and any attached files.

Reviewer #1: No

Reviewer #2: No

---

## [Author Response · Author response to Decision Letter 1]

22 Jun 2022

First of all, we would like to express our gratitude to both Referees for their help to improve the quality of our study. We discussed all criticisms and suggestions of the Referees and made changes accordingly. The suggestions are answered below point-by-point; textual changes are marked in the final manuscript.

Reviewer #1: The authors have adequately addressed all review comments. This work makes an interesting contribution to the field of equivalence learning and the role of sensory modalities (visual and multisensory) in various age groups.

A few minor English/grammar corrections to note:

Refer to lines 22, 25, 411, and 525 in the tracked version

Answer: The English/grammar corrections were performed with the help of a native English speaker.

Reviewer #2: The authors have largely addressed the points raised in the reviews, and the manuscript has been improved. However, I have one remaining comment:

Authors now report correlations between age and performance in each performance metric individually (i.e., separately for the V and the AV paradigms), and find, maybe not surprisingly, that performance generally increases with age. However, my original comment pertained to the correlation between age and the size of the multisensory gain (i.e., the performance difference between V and AV paradigms, e.g., MNAT minus VNAT). At least for NAT and ALER, the results presented in Tables 3 and 4 seem to suggest that the multisensory gain becomes smaller in >12 as compared to <12, and I wonder if there would be a significant correlation here? If yes, this would indicate that the multisensory gain actually decreases and becomes more adult-like over development, which would be different to the current interpretation that there is no difference until age 18.

Similarly, in Table 5 the authors did not directly test whether there is a significant difference between <12 and >12 in the multisensory gain, they only compare V and AV metrics separately between age groups. I think it would be crucial to directly compare the size of the multisensory gain between age groups (note that multisensory gain could be significant in both age groups as shown in Table 4 but still be significantly smaller in >12 than in <12).

Answer: We have defined the multisensory gain in Data Analysis section. We have calculated the gain values of each psychophysical learning parameters (NAT, ALER, RER, GER), and made the correlations of these parameters with age. Significant correlation of multisensory gain and age was found in acquisition phase. This indicates that the multisensory gain actually decreases with the aging of the children and becomes more adult-like over development. We have added a new paragraph “Correlation of multisensory gain with age and the size of multisensory gain in the age groups” to the results section. Two new tables were prepared (Table 6 and Table 7), which contain the descriptive statistic and the correlation results in detailed.

We have also compared the multisensory gains between the two age groups (under and above 12 years). The comparison of the multisensory gains of the two age cohorts denoted that the older cohort has smaller gain than the younger one, and the differences are significant in the acquisition phase. Table 8 contain the detailed results of the Mann-Whitney U tests.

We have additionally modified the Abstract and Discussion sections according to these new results.

---

## [Editor Report · Decision Letter 2]

4 Jul 2022

Multisensory Stimuli Enhance the Effectiveness of Equivalence Learning in Healthy Children and Adolescents

PONE-D-21-24459R2

Dear Dr. Eördegh,

We’re pleased to inform you that your manuscript has been judged scientifically suitable for publication and will be formally accepted for publication once it meets all outstanding technical requirements.

Kind regards,

Patrick Bruns

Guest Editor

PLOS ONE

Additional Editor Comments (optional):

I would like to disclose that I had previously served as a reviewer (Review #2) for this manuscript myself. Based on my assessment of the revised manuscript, I am happy to confirm that the remaining minor comments I had raised in the previous round of reviews have now all been addressed. Congratulations on a nice contribution.

Reviewers' comments:

None

---

## [Editor Report · Acceptance letter]

21 Jul 2022

PONE-D-21-24459R2 

Multisensory Stimuli Enhance the Effectiveness of Equivalence Learning in Healthy Children and Adolescents 

Dear Dr. Eördegh:

I'm pleased to inform you that your manuscript has been deemed suitable for publication in PLOS ONE. Congratulations! Your manuscript is now with our production department. 

Kind regards, 

on behalf of

Dr. Patrick Bruns 

Guest Editor

PLOS ONE